# QTL Mapping and Candidate Gene Analysis for Starch-Related Traits in Tartary Buckwheat (*Fagopyrum tataricum* (L.) Gaertn)

**DOI:** 10.3390/ijms25179243

**Published:** 2024-08-26

**Authors:** Juan Huang, Fei Liu, Rongrong Ren, Jiao Deng, Liwei Zhu, Hongyou Li, Fang Cai, Ziye Meng, Qingfu Chen, Taoxiong Shi

**Affiliations:** Research Center of Buckwheat Industry Technology, College of Life Science, Guizhou Normal University, Guiyang 550001, China; 201509007@gznu.edu.cn (J.H.); 18386012438@gznu.edu.cn (F.L.); 20010100382@gznu.edu.cn (R.R.); ddj613@163.com (J.D.); 201505005@gznu.edu.cn (L.Z.); lihongyouluod@163.com (H.L.); caifang919@gmail.com (F.C.); iorimouse@126.com (Z.M.); cqf1966@163.com (Q.C.)

**Keywords:** Tartary buckwheat (*Fagopyrum tataricum*), QTL mapping, amylose, amylopectin, total starch, candidate genes, SNP/InDel variations

## Abstract

Starch is the main component that determines the yield and quality of Tartary buckwheat. As a quantitative trait, using quantitative trait locus (QTL) mapping to excavate genes associated with starch-related traits is crucial for understanding the genetic mechanisms involved in starch synthesis and molecular breeding of Tartary buckwheat varieties with high-quality starch. Employing a recombinant inbred line population as research material, this study used QTL mapping to investigate the amylose, amylopectin, and total starch contents across four distinct environments. The results identified a total of 20 QTLs spanning six chromosomes, which explained 4.07% to 14.41% of the phenotypic variation. One major QTL cluster containing three stable QTLs governing both amylose and amylopectin content, *qClu-4-1*, was identified and located in the physical interval of 39.85–43.34 Mbp on chromosome Ft4. Within this cluster, we predicted 239 candidate genes and analyzed their SNP/InDel mutations, expression patterns, and enriched KEGG pathways. Ultimately, five key candidate genes, namely FtPinG0004897100.01, FtPinG0002636200.01, FtPinG0009329200.01, FtPinG0007371600.01, and FtPinG0005109900.01, were highlighted, which are potentially involved in starch synthesis and regulation, paving the way for further investigative studies. This study, for the first time, utilized QTL mapping to detect major QTLs controlling amylose, amylopectin, and total starch contents in Tartary buckwheat. The QTLs and candidate genes would provide valuable insights into the genetic mechanisms underlying starch synthesis and improving starch-related traits of Tartary buckwheat.

## 1. Introduction

With the development of society, our demand for crops has gradually shifted from meeting basic food needs to pursuing balanced nutrition. Some coarse cereals, such as Tartary buckwheat (*Fagopyrum tataricum* (L.) Gaertn), have gradually caught people’s attention because of their rich nutrition and health care functions, attracting great interest from researchers. Tartary buckwheat belongs to the genus *Fagopyrum* of *Polygonaceae* [1,2]. Tartary buckwheat has great market potential and development prospects for its high content of flavonoids, dietary fiber, vitamins, minerals, procyanidins, fatty acids, and other substances [3,4,5]. Flavonoids, which are lacking in grain crops such as rice and wheat, have the function of lowering blood sugar, blood lipids, and blood pressure, and they have antibacterial effects [6,7].

Starch is the major component of dietary carbohydrates, providing 40–60% of the energy for the human body. Based on chemical structure characteristics, starch can be divided into two types, namely amylose and amylopectin [8]. Amylose has a chain-like molecular structure without branches, while amylopectin has a branched molecular structure similar to a tree [9]. Their contents, the ratio of amylose to amylopectin, and the layered fine structure they formed determine the diversity of physicochemical properties of starch and the quality of grains [10].

The endosperm of cereals is the major site for starch synthesizing and storing. Storage starch is synthesized in the amyloplasts of the cereal endosperm using sucrose as the carbon source [8,10,11]. Glucose generated by photosynthesis in chloroplasts is converted into sucrose under the catalysis of sucrose phosphate synthase and then transported to the amyloplasts of storage tissues. UGPase catalyzes the hydrolysis of sucrose into fructose and UDPG, followed by the production of glucose-1-phosphate. Subsequently, ADP-glucose pyrophosphorylase catalyzes the synthesis of ADPG from glucose-1-phosphate. Finally, through a series of enzymatic reactions under the action of enzymes such as granule-bound starch synthase (GBSS), soluble starch synthase (SS), starch branching enzyme (SBE), and debranching enzyme (DBE), ADPG is synthesized into amylose and amylopectin.

Up to date, numerous genes encoding enzymes for starch biosynthesis have been identified and cloned, including *Wx*, *SSIII*, *SBE*, *ISA*, etc. The *Wx* gene is the primary gene regulating amylose synthesis in rice, and it has multiple allelic variants corresponding to different amylose content classes, including *wx*, *Wx^op^*, *Wx^b^*, *Wx^in^*, *Wx^a2^*, and *Wx^a1^* [12]. The *SSIII*, *SBE*, and *ISA* genes are the major genes for amylopectin synthesis, and their mutation leads to a reduced content of amylopectin [13,14]. Transcription factors (TFs), such as NAC, ZIP, ERF, and WRKY family TFs, can bind to the promoter of the starch biosynthesis genes and thus regulate the pathway of starch biosynthesis [15,16].

The quantitative trait locus (QTL) for starch-related traits has been reported in multiple species, including wheat, rice, and corn. For example, QTL mapping amylopectin content and amylose content was performed on wheat grain, resulting in one and four major QTLs detected for amylopectin and amylose content, respectively [17]. Guo et al. evaluated the amylose content, the gel consistency, the gelatinization temperature, and the protein content, resulting in a total of seven QTLs and twelve pairs of QTLs detected to have significant additive and epistatic effects for these four traits in rice [18]. Hu et al. comprehensively analyzed the genetic basis of maize grain starch content via single linkage mapping, joint linkage mapping, and a genome-wide association study. The result showed fifty unique single QTLs for starch content with eighteen novel QTLs detected, among which only five QTLs explained over 10% phenotypic variation in single populations [19].

Starch accounts for 60% to 70% of the Tartary buckwheat grain, among which amylose content ranges from 10% to 28% [20]. In recent years, genes in the starch biosynthesis process in Tartary buckwheat were gradually studied. *FtGBSSI*, involved in amylose biosynthesis and composed of 14 exons and 13 introns, was the first isolated gene in Tartary buckwheat [21]. Our previous transcriptome analyses illustrated several starch biosynthetic and regulatory genes and proposed two potential gene regulatory networks involved in starch biosynthesis [22,23]. Two *GBSS* genes (*FtGBSSII-4* and *FtGBSSII-5*) and three *SSS* genes (*FtSSII-2*, *FtSSIII-1*, and *FtSSIV-2*) were mainly expressed in seeds, indicating that they may be the dominant isoforms for starch synthesis in Tartary buckwheat [24]. However, there are no reports on the QTL mapping of starch-related traits in Tartary buckwheat at present. In this study, we determined the amylose content, amylopectin content, and total starch content using a Tartary buckwheat recombinant inbred lines (XJ-RILs) population [25]. QTL mapping was performed, and candidate genes related to starch within the major QTLs were predicted by combining analyses of SNP/InDel mutations, expression patterns, and enriched KEGG pathway. This work is crucial to excavate key genes involved in starch synthesis and regulation and promote the molecular breeding of high-quality starch in Tartary buckwheat.

## 2. Results

### 2.1. Variation Analysis for Starch-Related Traits

We examined the content of starch-related traits, including amylose, amylopectin, and total starch, in two parents (Jinqiaomai2 and Xiaomiqiao) and the XJ-RILs population under the three environments (2021S, 2021A, and 2022S) (Table 1). The ANOVA analysis showed that the amylose, amylopectin, and total starch content of Xiaomiqiao were significantly higher than that of Jinqiaomai2 in all three environments (*p* < 0.05), except for the amylopectin content in 2022S. The average amylose content in the XJ-RILs population under the three environments was 19.26%, 19.76%, and 22.39%, respectively. The average amylopectin content under the three environments was 49.83%, 32.09%, and 40.46%, respectively. The average total starch content under the three environments was 69.08%, 51.84%, and 62.85%, respectively.

The variation range of amylose content in the XJ-RILs population was 11.19–23.45%, 15.70–22.50%, and 17.20–28.02%, with mean values of 19.26%, 19.76%, and 22.39% under the three environments, respectively (Table 1). The variation range of amylopectin content in the XJ-RILs population was 36.80–59.68%, 23.66–39.93%, and 30.26–51.61%, with mean values of 49.83%, 32.09%, and 40.46% under the three environments, respectively. The variation range of total starch content in the XJ-RILs population was 57.73–78.72%, 45.61–61.52%, and 53.07–74.15%, with mean values of 69.08%, 51.84%, and 62.85% under the three environments, respectively. The coefficients of variation (CV) of the three traits ranged from 5% to 11% under the three environments, among which the CV of total starch content was the smallest and that of amylopectin content was the largest.

The three traits in the XJ-RILs population showed continuous variation and significant transgressive segregation under the three environments (Figure 1). Each trait produced a continuous, single peak and skewed distribution pattern (Table 1, Figure 1). The absolute values of skewness and kurtosis of amylose, amylopectin, and total starch content under the three environments were less than 1.0, except for total starch content in 2021A (Table 1). Meanwhile, their frequency distribution showed a clear main peak (Figure 1). These indicated the amylose, amylopectin, and total starch content under the three environments were generally normally distributed, which is consistent with the characteristics of quantitative traits.

The correlation analysis revealed a significant negative correlation between amylose and amylopectin content in all environments (*p* < 0.05 or 0.01). Amylopectin and total starch content showed a significant positive correlation in all environments (*p* < 0.01). Amylose and total starch content showed a significant positive correlation only in the 2022S environment. Among them, the correlation coefficient between amylopectin and total starch was the highest (more than 0.881) (Table 2).

### 2.2. QTL Mapping of Starch-Related Traits

QTL mapping was performed based on phenotypic data of the XJ-RILs population and the ultrahigh-density genetic map constructed previously [25]. As a result, a total of 20 QTLs were detected for amylose, amylopectin, and total starch on chromosomes Ft1, Ft2, Ft3, Ft4, Ft5, and Ft7 under the four environments (2021S, 2021A, 2022S, and BLUP). These QTLs were distributed on chromosomes Ft1, Ft2, Ft3, Ft4, Ft5, and Ft7 and explained 4.07–14.41% of the phenotypic variation, with LOD scores ranging from 2.54 to 7.37 (Table 3, Figure 2). Of these QTLs, four were major QTLs with a phenotypic variation greater than 10%.

Five QTLs for amylose content were located on chromosomes Ft3, Ft4, and Ft5, with the phenotypic variation ranging from 5.08% to 11.04%. Among these, *qAL4-1,* located in the physical interval of 39.85~42.74 Mbp on the chromosome Ft4, was repeatedly detected across two environments, explaining 6.46–9.52% of the phenotypic variation. *qAL4-2*, located in the physical interval of 42.10~43.34 Mbp on the chromosome Ft4, was repeatedly detected across three environments and showed the greatest effects of all five QTLs, explaining 5.08–10.12% of the phenotypic variation. The additive effects of *qAL4-1* and *qAL4-2* were both derived from Jinqiaomai2.

Eleven QTLs for amylopectin content were distributed on chromosomes Ft1, Ft2, Ft3, Ft4, and Ft7, with the phenotypic variation ranging from 4.07% to 14.41%. Among these QTLs, *qALP4-3*, located in the physical interval of 39.85~42.65 Mbp on the chromosome Ft4, was repeatedly detected across three environments and showed the greatest effects, explaining 8.59–14.41% of the phenotypic variation. The additive effects of *qALP4-3* were derived from Xiaomiqiao.

Four QTLs for total starch content were located on chromosomes Ft1, Ft4, and Ft7, with the phenotypic variation ranging from 4.41% to 7.99%. Among these, *qTSC7-1*, located in the physical interval of 35.85~40.63 Mbp on the chromosome Ft7, was repeatedly detected in two environments and showed the greatest effects, explaining 4.41–7.99% of the phenotypic variation. The additive effects of *qTSC7-1* were derived from Jinqiaomai2.

### 2.3. QTL Cluster Analysis of Starch-Related Traits

The 20 QTLs for the starch-related traits were located on 15 chromosomal regions. Several QTLs of different traits exhibited an overlapping confidence interval, which formed three QTL clusters (Table 4). Among them, *qClu-4-1*, located in the physical interval of 39.85~43.34 Mbp on chromosome Ft4, was a major QTL cluster composed of three stable QTLs (*qALP4-3*, *qAL4-1*, and *qAL4-2*) repeatedly detected at least in two environments. *qClu-1-1*, located in the physical interval of 6.45–9.15 Mbp on chromosome Ft1, was formed by three environment-specific QTLs (*qALP1-1*, *qTSC1-1*, and *qALP1-2*). *qClu-7-1*, located in the physical interval of 35.85~40.63 Mbp on chromosome Ft7, was composed of one stable QTL repeatedly identified in two environments (*qTSC7-1*) and one environment-specific QTL (*qALP7-2*).

### 2.4. Identification of Candidate Genes within the Major QTL Cluster qClu-4-1

Notably, cluster *qClu-4-1* included three stable QTLs controlling amylose and amylopectin content. Hereafter, the physical intervals of *qClu-4-1* were aligned to the current reference genome of Tartary buckwheat [26], resulting in 239 candidate genes identified (Appendix A). Of these, two hundred seventeen candidate genes were expressed in at least one tissue among root, stem, leaf, flower, and seed, and they were divided into six clusters (C1–C6) based on their expression patterns (Figure 3a). C1 included 39 genes that were highly expressed in the root, moderately expressed in the stem, leaf, and flower, and lowly expressed in seeds. C2 included 50 genes that were highly expressed in root, stem, and leaf and lowly expressed in the flower and seeds. C3 included 36 genes that were highly expressed in the stem and leaf, moderately expressed in the flower, and lowly expressed in the root and seeds. C4 included 21 genes that were lowly expressed in the root and stem and moderately expressed in the leaf, flower, and seed. C5 included 36 genes that were highly expressed in root, stem, leaf, and seed, whereas lowly expressed in flower. C6 included 35 genes that were highly expressed in seed, moderately expressed in root, and lowly expressed in stem, leaf, and flower.

Genes expressed in the seed seem more likely to be associated with grain starch-related traits. Among these annotated genes, 92 genes in C4, C5, and C6 clusters showed higher expression levels in seed. Based on KEGG analysis, five pathways, namely starch and sucrose metabolism, metabolic pathways, cysteine and methionine metabolism, biotin metabolism, and biosynthesis of secondary metabolites, were significantly enriched (*p* < 0.05, Figure 3b). Three genes, FtPinG0005109900.01, FtPinG0005125200.01, and FtPinG0004899600.01, were included in the pathway of starch and sucrose metabolism. FtPinG0005109900.01 was a homolog to *Arabidopsis starch synthase 2* (*AtSS2*), which was essential for amylopectin biosynthesis. FtPinG0005125200.01 was homologues to *Arabidopsis sucrose synthase 4* (*AtSUS4*). FtPinG0004899600.01 was a homolog to *Arabidopsis alpha-glucan phosphorylase 2* (*AtPHS2*).

### 2.5. Analysis of the SNP/InDel Variations of Candidate Genes for qClu-4-1

Comparative genomics analysis between the two parents was carried out to identify candidate genes with the SNP/InDel variations within the physical intervals of the stable major QTL cluster *qClu-4-1*. Totally, 87 annotated genes located within the physical intervals of *qClu-4-1* have SNP/InDel variations between the parent Xiaomiqiao and Jinqiaomai2 (Appendix A), of which 28 genes belonged to C4, C5, or C6 clusters showed higher expression level in seeds compared with other tissues. Among these twenty-eight genes, nine have yet to be functionally characterized, while the remaining nineteen genes showed a total of fifty-nine SNP/InDel variations (Table 5).

FtPinG0004897100.01, annotated to rice *SHR*, exhibited six SNP/Indel mutations in its 5’ untranslated region (UTR) or 3′ UTR. FtPinG0002636200.01, annotated to glycosyltransferase family protein, exhibited 1 SNP mutation in 3’ UTR. FtPinG0009329200.01, annotated to transporter family protein, exhibited 1 SNP mutation in its intron. FtPinG0007371600.01, annotated to phosphatidylinositol-4-phosphate 5-kinase family protein, exhibited 1 SNP mutation in 3′ UTR. FtPinG0005109900.01, annotated to starch synthase family protein, exhibited 1 SNP mutation in 3’ UTR.

The SNP/InDel variations between two parents of FtPinG0004897100.01, FtPinG0002636200.01, FtPinG0009329200.01, FtPinG0007371600.01, and FtPinG0005109900.01 were verified by the Sanger sequence (Figure 4). A total of five out of the six SNP/InDel mutations in FtPinG0004897100.01 were verified. The first mutation was an SNP mutation in the 5′ UTR of this gene, whose base was “C” in Jinqiaomai2, whereas “T” in Xiaomiqiao (Figure 4A). The second mutation was an InDel mutation in the 5′ UTR, whose base was “AACAAC” in Jinqiaomai2, whereas “AAC” in Xiaomiqiao (Figure 4B). The third mutation was an SNP mutation in the 3′ UTR, whose base was “T” in Jinqiaomai2, whereas “C” in Xiaomiqiao (Figure 4C). The fourth and fifth mutations were SNP mutations in the 3′ UTR, whose base was “G” in Jinqiaomai2, whereas “A” in Xiaomiqiao of both sites (Figure 4D). One SNP mutation was verified in the 3′ UTR of FtPinG0002636200.01, whose base was “A” in Jinqiaomai2, whereas “T” in Xiaomiqiao (Figure 4E). One SNP mutation was verified on the intron of FtPinG0009329200.01, whose base was “C” in Jinqiaomai2, whereas “G” in Xiaomiqiao (Figure 4F). Three consecutive SNP mutations were verified on the 3′ UTR of FtPinG0007371600.01, whose base was “GGG” in Jinqiaomai2, whereas “AAT” in Xiaomiqiao (Figure 4G). However, the SNP mutation in FtPinG0005109900.01 was not identified by the Sanger sequence.

### 2.6. Starch Content and Expression of Candidate Genes in Two Parents

We analyzed the starch content between the two parents of the XJ-RILs population, and the results revealed that the amylose, amylopectin, and total starch content of Jinqiaomai2 were higher than those of Xiaomiqiao (Figure 5A). Concurrently, the expression levels of four genes that were predominantly expressed in the seeds and exhibited SNP/InDel variations and one gene annotated to rice starch synthase were analyzed between the two parents (Figure 5B). The expression levels of FtPinG0004897100.01, FtPinG0002636200.01, and FtPinG0009329200.01 in Jinqiaomai2 were lower than those in Xiaomiqiao, which were contrary to the trend of starch content, suggesting that they were negatively regulated genes. Conversely, the expression level of FtPinG0007371600.01 in Jinqiaomai2 was higher than that in Xiaomiqiao, which showed a similar trend of starch content, indicating that it was a positively regulated gene. However, the expression of FtPinG0005109900.01 showed no obvious difference between the two parents.

## 3. Discussion

Starch content is a quantitative trait controlled by multiple genes and influenced by both genetic and environmental factors. QTL mapping is a good method for studying quantitative traits. However, no one has yet used QTL mapping to study starch traits in Tartary buckwheat. Using the XJ-RILs population as the material, this study utilizes the advantages of QTL mapping to detect the amylose, amylopectin, and total starch contents under the three environments. Based on the high-density SNP genetic linkage map constructed for the XJ-RILs population, QTL mapping was performed for starch-related traits. The results detected a total of 20 QTLs for amylose, amylopectin, and total starch on six chromosomes and explained 4.07–14.41% of the phenotypic variation. Three QTL clusters were formed, among which *qClu-4-1* included three stable QTLs controlling amylose and amylopectin. A total of 239 candidate genes were predicted in this QTL cluster, and their SNP/InDel mutation, expression patterns, and enriched KEGG pathway were analyzed. Ultimately, five key candidate genes (FtPinG0004897100.01, FtPinG0002636200.01, FtPinG0009329200.01, FtPinG0007371600.01, and FtPinG0005109900.01) that may be involved in starch synthesis and regulation were predicted.

The variation ranges of amylose, amylopectin, and total starch contents of 221 RILs were 11.19–28.02%, 23.66–59.68%, and 45.61–78.72%, respectively. This indicates that the starch content in t Tartary buckwheat grain exhibits abundant genetic and environmental variation, which were consistent with previous reports that the amylose content ranges from 10% to 28%, and the total starch content is approximately 70% [6,20]. In addition, three starch-related traits showed an approximated normal distribution and expressed transgressive segregation under different environments, which falls under the characteristics of quantitative traits [27,28]. Additionally, the amylopectin and total starch contents were significantly lower in the autumn of 2021 compared to the spring of 2021 and 2022. This suggests that the environment has an extreme influence on amylopectin. Other studies have already discovered and proven this phenomenon. For example, Rhazi et al. investigated the genetic and environmental variation in starch content and characteristics of 14 bread wheat cultivars. The results indicated that genetics, environment, and their interaction had significant effects on starch content, starch granules distribution, the percentage of amylose and amylopectin, as well as their molecular characteristics, including weight-average molar mass, number-average molar mass, polydispersity, and gyration radius [29]. High-light-induction was identified as a promising method for promoting starch accumulation in duckweed. Duckweed was cultivated under different photoperiods and light intensities, and it was found that the biomass and starch production of duckweed increased with an increase in light intensity and photoperiod [30]. Spring and autumn have different light intensities and photoperiods. The photoperiod is longer in spring than in autumn, which could partly explain why the amylopectin and total starch contents were higher in spring than in autumn. From 221 RILs, we can select lines that exhibit significant differences in amylose, amylopectin, and total starch content across three different environments and recommend them as germplasm resources with high amylose, high amylopectin, and high starch content for Tartary buckwheat breeding. Numerous QTLs and candidate genes for starch-related traits have been identified in multiple species, including wheat, rice, and corn [17,18,19]. Here, we reported the first case that used QTL mapping to study starch-related traits in Tartary buckwheat. We identified 5, 11, and 4 QTLs for amylose, amylopectin, and total starch content, respectively. Quantitative traits are greatly influenced by the environment, such as light, temperature, precipitation, and soil conditions, which can affect the expression and regulation of related genes. Therefore, overlapping QTLs detected under multiple environments, including multiple years and multiple locations, are much more reliable. Two QTLs for amylose content (*qAL4-2* and *qAL4-1*) on the chromosome Ft4, one QTL for amylopectin content (*qALP4-3*), and one QTL for total starch content (*qTSC7-1*) were repeatedly detected in at least two environments, indicating that these four QTLs are stable QTLs that control starch-related traits in Tartary buckwheat. In addition, three (*qALP4-3*, *qAL4-1*, and *qAL4-2*) out of these four stable QTLs are located within a physical interval of 39.85~43.34 Mbp on chromosome 4 and are grouped into a QTL cluster (*qClu-4-1*), which is a major region controlling starch-related traits. In our previous study, QTL for thousand-grain weight (TGW)s was detected on Chromosome Ft1 and Chromosome Ft4 [25,31]. Coincidentally, the confidence interval of a stable QTL controlling TGW on Chromosome Ft4 ranged from 120.1 to 132.3 cM, overlapped with the confidence interval (120.3~136.5 cM) of *qClu-4-1* controlling starch-related traits. As the most abundant component in crop grains, starch content directly affects crop yield. For Tartary buckwheat, starch accounts for approximately 70% of the total weight of grain, which is significantly positively correlated to grain yield [32]. *qClu-4-1* is located in the same region as the QTL controlling TGW in Tartary buckwheat, further confirming the above results and indicating that the QTL we have identified is highly reliable. Furthermore, *qClu-4-1* can provide a basis for further fine-mapping of starch-related traits and the development of molecular markers that are tightly linked to starch traits.

By integrating the results of parental re-sequencing and tissue-specific expression, 239 genes located in the interval of *qClu-4-1* were analyzed in depth. Of them, five genes, namely FtPinG0004897100.01, FtPinG0002636200.01, FtPinG0009329200.01, FtPinG0007371600.01, and FtPinG0005109900.01, exhibited not only SNP/InDel variations but also high expression levels in seeds and were identified as key candidate genes involving in the starch-related traits of Tartary buckwheat grains.

Among the above five key candidate genes, FtPinG0004897100.01 is annotated to rice *SHR*, which belongs to the GRAS family TF. One of these TF family members, ZmGRAS20, has been reported to function as a starch biosynthesis regulator. *ZmGRAS20* has very high expression levels in developing corn grains and is specifically expressed only in the endosperm [33]. Overexpression of *ZmGRAS20* in rice leads to a chalkiness characteristic with an altered starch content and structure [34]. In sorghum, GRAS family TFs were expressed abundantly in the developmental stage of grains, which suggests they might be involved in the biosynthesis of starch and phenolic compounds [35]. FtPinG0004897100.01 is highly expressed in seeds rather than other tissues and has five SNP/InDel mutations in the 5’ UTR or 3’ UTR, which suggests FtPinG0004897100.01 as a potential regulator in starch biosynthesis in Tartary buckwheat.

Another key candidate gene, FtPinG0005109900.01, is annotated to rice *starch synthase* (*SS*), which encodes an essential enzyme in amylopectin biosynthesis of cereals [11,13]. SS has the largest number of isoforms and is divided into five subclasses, namely SSI, SSII, SSIII, SSIV, and SSV [11,36]. Among them, SSI, SSII, and SSIII are generally responsible for a-glucan chain elongation, whereas SSIV and SSV are generally responsible for starch granule initiation [36,37]. For FtPinG0005109900.01, the SNP mutation was not verified, and there was no obvious difference in its expression between the parents. One possible reason is that it might not be the key gene regulating amylopectin synthesis in Tartary buckwheat. However, another possible reason for this phenomenon is that it exhibits post-transcriptional regulation, such as post-translational modifications or protein–protein interactions. It is reported that multiple enzymes involved in amylopectin synthesis, including SSI, SSIIa, SSIIIa, SBEIIb, SBEI, and ISA2, undergo phosphorylation modifications in rice [11]. Analysis of starch granule-associated phosphoproteins in wheat also gives us evidence that SBEII and two SS forms are phosphorylated [38]. In maize endosperm, GBSS, SSI, SSIIa, BEIIb, SSIII, BEI, BEIIa, and starch phosphorylase have been identified as granule-bound internal proteins, among which directly protein–protein interactions exhibit, and the abundance of one protein can be influenced by another amylopectin biosynthetic enzyme [39]. The above provides direct evidence that enzymes of amylopectin synthesis are regulated by protein phosphorylation and protein–protein interactions. Therefore, further experiments are needed to verify whether protein phosphorylation exists in amylopectin-related enzymes, such as FtPinG0005109900.01, in Tartary buckwheat.

It is worth mentioning that it is relatively preliminary to verify the function of five key candidate genes only based on the SNP/InDel variations and gene expression analyses. More in-depth functional studies combining biochemistry and molecular biology techniques, such as enzyme activity assay, transgenic experiments, and gene regulatory network identification, are important and need to be carried out to study the specific roles of these genes and their encoded proteins in the following step. These would provide us with excellent gene resources for molecular breeding of Tartary buckwheat.

## 4. Materials and Methods

### 4.1. Materials Planting and Sampling

The XJ-RILs populations containing 221 lines derived from the cross of “Xiaomiqiao” (female parent) and “Jinqiaomai2” (male parent) were used as the plant materials in this study [25]. Using randomized block design, the XJ-RILs population and their parents were planted in Baiyi (26°64′ N, 106°63′ E) of Guizhou Province in the Spring of 2021 and Anshun (26°17′ N, 106°18′ E) of Guizhou Province in the Autume of 2021 and Spring of 2022. Each line was planted in three blocks; each block was 2 m long, with three rows 33 cm apart. The field was given with conventional field management. At the seedling stage, new leaves were taken and stored in a refrigerator at −20 °C for DNA extraction. At maturity stage, fully mature seeds on the main stems were collected. Three replicates were taken from each line. They were stored in a 4 °C refrigerator for the measurement of starch content.

### 4.2. Measurement and Analysis of the Content of Starch-Related Traits

Tartary buckwheat grain of XJ-RILs and their parents were dried at a constant temperature of 60 °C for three days, hulled, ground, and then sieved through a 100-mesh sieve. Amylose and amylopectin content were measured using our previous method [23]. Total starch content is equal to the sum of amylose and amylopectin content. Three technical replicates were taken for each sample. Descriptive statistics for the starch-related traits were analyzed using Excel version 2019 software. Descriptive statistical analysis and correlation analysis were analyzed using IBM SPSS Statistics version 22.

### 4.3. QTL Mapping

QTL analysis was performed as our previous method with slight modifications [25,31]. In short, a high-density genetic map with 4151 markers and 122,185 SNP markers that we previously constructed for the XJ-RIL population was used as a genotype [25]. QTL mapping was performed by Windows QTL Cartographer, version 2.5, with a composite interval mapping model. The empirical threshold for CIM was set as 1000 permutations, and QTL was determined to be LOD > 2.5. QTLs for the same trait from different environments with confidence intervals overlapping and the additive effects originating from the same parental line were identified as the same QTL. A QTL that explained more than 10% of phenotypic variation in at least one environment was considered a major QTL [25,31,40]. QTLs for different traits with overlapping confidence intervals were considered as a QTL cluster [31].

### 4.4. Candidate Gene Prediction and Analyses

Genomic data of Tartary buckwheat [26] were used to identify candidate genes in the major QTL cluster. The candidate genes were annotated by the local BLASTP program (ncbi-blast-2.11.0+) using the protein sequences as query and rice genomic protein sequences (http://rice.uga.edu/, accessed on 1 August 2024) as a database with an e-value of 0.00001 and max_target_seqs 1. The KEGG pathway was annotated and enriched by KOBAS-I (http://bioinfo.org/kobas, accessed on 1 August 2024) [41] with an e-value of 0.00001. Expression data for candidate genes in different tissues were retrieved from the transcriptome of Tartary buckwheat genome data [26]. Expression data for candidate genes in the seeds of Jinqiaomai2 and Xiaomiqiao were retrieved from our previous transcriptome data [23]. Heatmap and enriched KEGG pathways were visualized in R language, version 4.4.0 (2024-04-24 ucrt) [42,43].

### 4.5. Verification of the SNP/InDel Variations of Candidate Genes

Genomic DNA was extracted using the plant total DNA extraction kit from Sangon Biotech Co., Ltd. (Shanghai, China). Primers were designed based on the sequences flanking the SNP/InDel variations of candidate genes using Primer Premier5 software. The primers were listed in Appendix A. PCR was performed using PrimeSTAR^®^ Max DNA Polymerase from Takara Biomedical Technology Co., Ltd. (Beijing, China). PCR products with a single band and correct fragment size were sent directly to Sangon Biotech Co., Ltd. (Shanghai, China) for Sanger sequencing.

## 5. Conclusions

Overall, we detected a total of 20 QTL for starch-related traits on six chromosomes and explained 4.07–14.41% of the phenotypic variation in Tartary buckwheat in this study. A stable major, *qClu-4-1*, included three stable QTLs controlling amylose and amylopectin on chromosome Ft4. Within its physical interval, 239 candidate genes were identified, among which FtPinG0004897100.01, FtPinG0002636200.01, FtPinG0009329200.01, FtPinG0007371600.01, and FtPinG0005109900.01 were predicted as key candidate genes involved in starch synthesis and regulation of Tartary buckwheat, based on the SNP/InDel variations and gene expression analyses. These results would enhance our understanding of the molecular genetic regulation of starch synthesis in Tartary buckwheat.

## Figures and Tables

**Figure 1 ijms-25-09243-f001:**
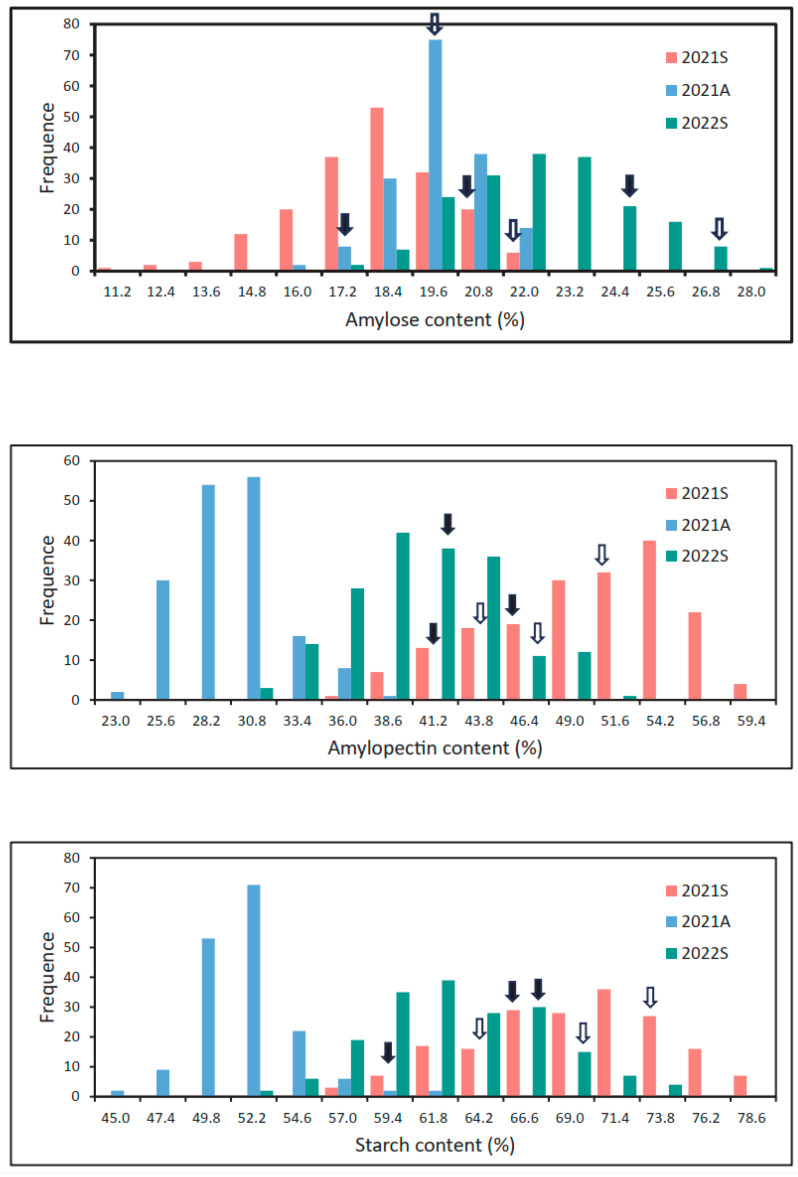
Frequency distribution of starch-related traits in the XJ-RILs population. The black solid arrow and hollow arrow represent the values of starch-related traits in Jinqiaomai2 and Xiaomiqiao, respectively.

**Figure 2 ijms-25-09243-f002:**
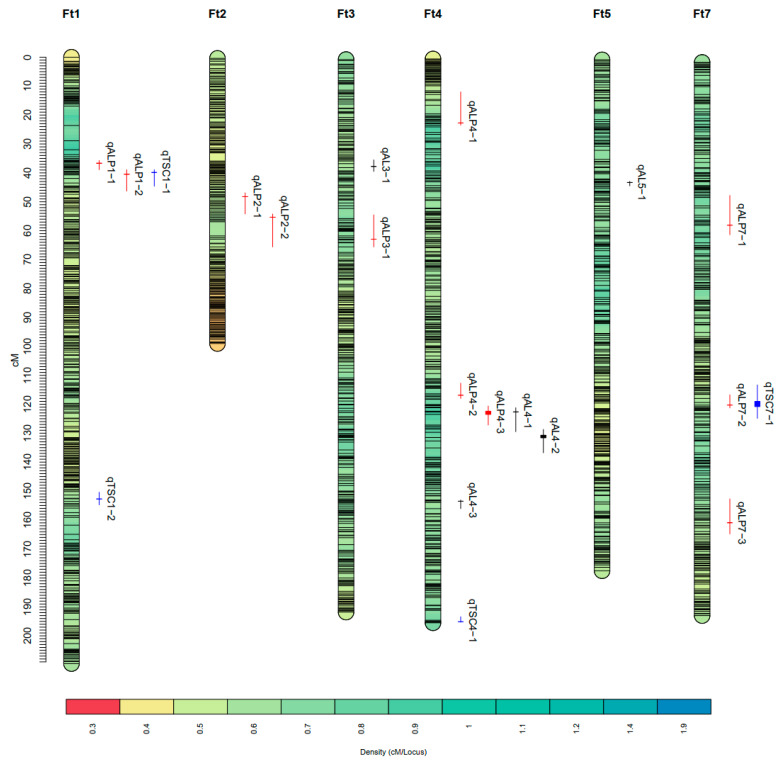
The distribution of the QTLs for amylose (AL), amylopectin (ALP), and total starch content (TSC) identified in the XJ-RILs population under the four environments. The black, red, and blue lines represent QTLs for amylose, amylopectin, and total starch, respectively. The horizon lines and vertical lines represent the peak position and the confidence interval of QTLs, respectively.

**Figure 3 ijms-25-09243-f003:**
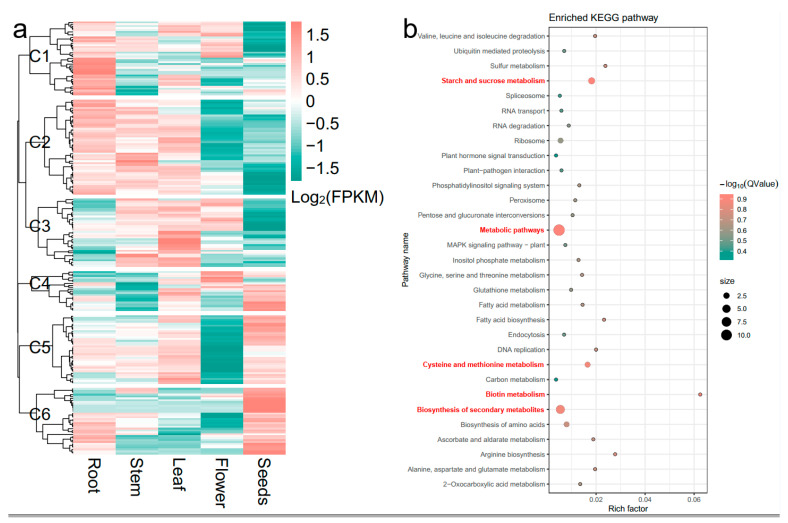
Identification of candidate genes in cluster *qClu-4-1*. (**a**) Expression patterns of candidate genes identified in cluster *qClu-4-1*. (**b**) KEGG enrichment of candidate genes in clusters C4, C5, and C6. Names in red font represent the significantly enriched pathways (*p* < 0.05).

**Figure 4 ijms-25-09243-f004:**
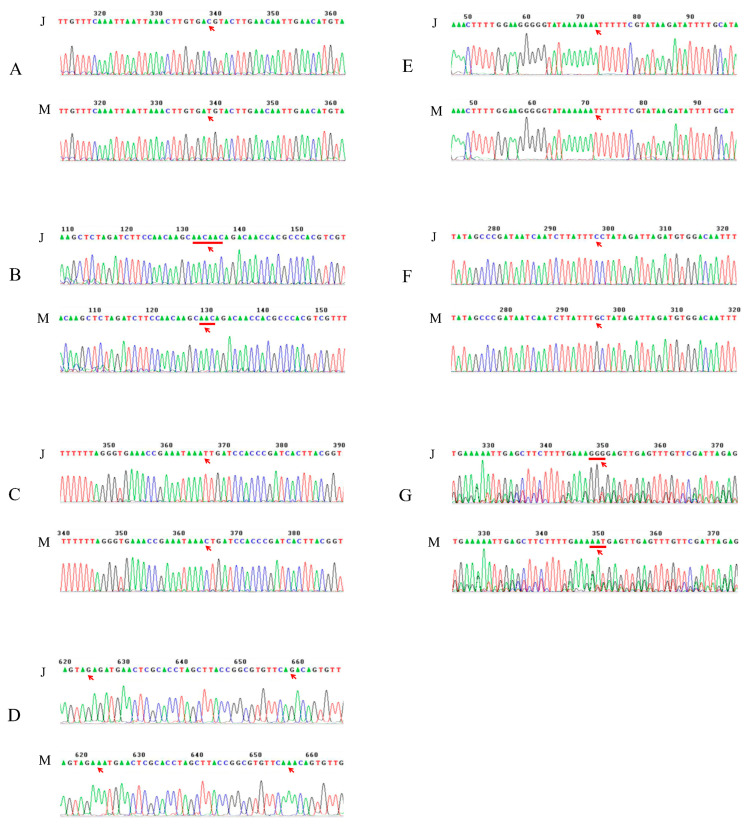
Verification of the SNP/InDel variations in candidate genes between two parents. (**A**–**D**), the SNP/InDel variations in FtPinG0004897100.01. (**E**), the SNP variation in FtPinG0002636200.01. (**F**), the SNP variation in FtPinG0009329200.01. (**G**), the InDel variation in FtPinG0007371600.01. J and M represents Jinqiaomai2 and Xiaomiqiao, respectively. Red arrows represent the sites of SNP/InDel variations.

**Figure 5 ijms-25-09243-f005:**
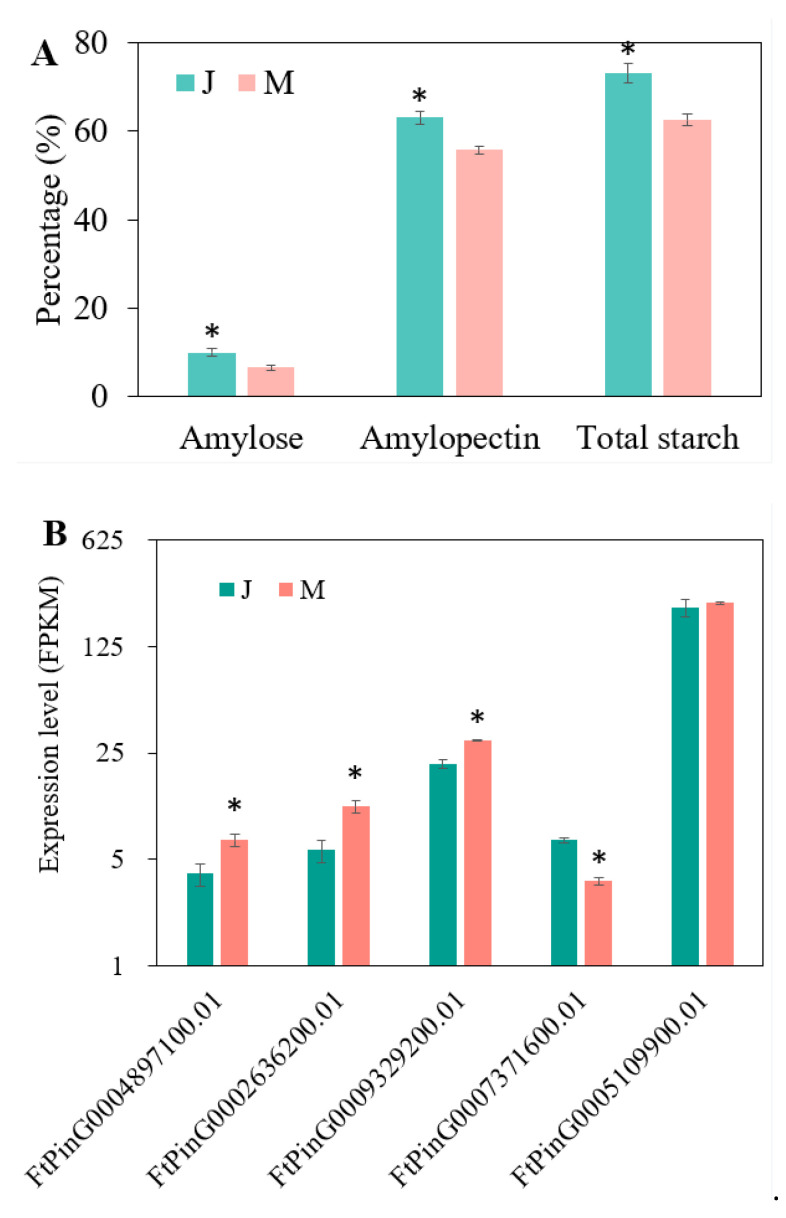
Starch content and expression of candidate genes in two parents. (**A**), Amylose, amylopectin, and total starch content of two parents. (**B**), Expression of candidate genes in two parents. J and M represents Jinqiaomai2 and Xiaomiqiao, respectively. The error bars indicate the SD values, while the asterisks (*) indicate the contents of starch-related traits or the expression level of genes between two parents has reached a statistically significant difference (*p* < 0.05).

**Table 1 ijms-25-09243-t001:** Descriptive statistical results for starch-related traits of the parents and XJ-RIL populations in three environments.

Traits	Environments	Parents		XJ-RILs Population				
		Jinqiaomai2	Xiaomiqiao	Mean ± SD	Range	Kurtosis	Skewness	CV (%)
Amylose (%)	2021S	20.90 ± 0.24 b	22.28 ± 0.91 a	19.26 ± 1.98	11.19~23.45	0.375	−0.528	10%
	2021A	17.80 ± 0.33 b	19.44 ± 0.61 a	19.76 ± 1.20	15.70~22.50	0.449	0.031	6%
	2022S	24.97 ± 0.64 a	26.19 ± 0.61 b	22.39 ± 2.15	17.20~28.02	−0.238	0.004	10%
Amylopectin (%)	2021S	47.03 ± 1.22 b	50.95 ± 1.87 a	49.83 ± 5.20	36.80~59.68	−0.442	−0.310	10%
	2021A	41.65 ± 1.69 a	43.61 ± 1.24 b	32.09 ± 2.86	23.66~39.93	0.634	0.632	9%
	2022S	42.98 ± 1.62 a	43.57 ± 1.08 a	40.46 ± 4.32	30.26~51.61	0.013	0.208	11%
Total Starch (%)	2021S	67.92 ± 1.17 b	73.24 ± 1.85 a	69.08 ± 4.94	57.73~78.72	−0.492	−0.178	7%
	2021A	59.45 ± 1.50 a	63.05 ± 1.62 b	51.84 ± 2.54	45.61~61.52	2.419	0.970	5%
	2022S	67.96 ± 1.49 a	69.76 ± 1.38 b	62.85 ± 4.46	53.07~74.15	−0.201	0.476	7%

CV, Coefficient of variation. 2021S, 2021A, and 2022S represent the Spring of 2021, the Autumn of 2021, and the Spring of 2022, respectively. The lowercase letters (a and b) represent a difference that is significant at the 0.05 level between the two parents.

**Table 2 ijms-25-09243-t002:** Pearson’s correlation analysis of starch-related traits in XJ-RILs population.

Trait	Environment	Total Starch	Amylopectin
Amylose	2021S	0.065	−0.315 **
	2021A	−0.016	−0.425 **
	2022S	0.307 **	−0.180 *
Amylopectin	2021S	0.927 **	
	2021A	0.912 **	
	2022S	0.881 **	

* and ** indicate significant correlation at the 0.05 and 0.01 levels, respectively.

**Table 3 ijms-25-09243-t003:** QTL analysis for three starch-related traits detected in the XJ-RILs population.

NO.	Trait	QTL	Environment	Chromosome	Position (cM)	LOD	R^2^ (%)	Additive Effect	Confidence Interval (cM)	Flanking Markers	Physical Interval (Mbp)
1	Amylose	*qAL3-1*	2021S	Ft3	37.51	2.68	5.2	−0.51	35.2~39.2	Block5303~Block5318	11.8~12.55
2	Amylose	*qAL4-1*	2021A	Ft4	122.31	3.31	6.46	0.55	120.9~129.2	Block8744~Block8836	39.85~42.74
			BLUP	Ft4	122.31	5.82	9.52	0.002			
3	Amylose	*qAL4-2*	2021S	Ft4	130.41	4.8	10.12	0.65	128.4~136.5	Block8830~Block8844	42.10~43.34
			2022S	Ft4	131.31	2.58	5.08	0.49			
			BLUP	Ft4	131.31	4.42	7.34	0.002			
4	Amylose	*qAL4-3*	2021A	Ft4	153.21	2.7	6.22	−0.34	152.8~155.8	Block9021~Block9046	46.58~48.92
5	Amylose	*qAL5-1*	2022S	Ft5	43.31	4.74	11.04	−0.49	42.9~44.5	Block9728~Block9753	8.39~9.06
6	Amylopectin	*qALP1-1*	BLUP	Ft1	36.61	5.31	8.09	−0.17	35.7~38.9	Block320~Block337	6.47~6.10
7	Amylopectin	*qALP1-2*	2022S	Ft1	40.51	3.54	6.33	−1.13	38.9~46.3	Block334~Block428	6.45~9.15
8	Amylopectin	*qALP2-1*	2021A	Ft2	48.21	4.09	9.04	2.15	46.9~54.2	Block2872~Block2992	15.34~21.46
9	Amylopectin	*qALP2-2*	2022S	Ft2	55.31	3.47	7.73	1.96	54.2~65.6	Block3073~Block3468	12.57~31.04
10	Amylopectin	*qALP3-1*	2021S	Ft3	62.61	3.37	7.22	−0.79	54.2~65.3	Block5579~Block5697	16.07~26.81
11	Amylopectin	*qALP4-1*	2021S	Ft4	22.41	2.97	5.52	−1.24	11.7~23.2	Block7563~Block7612	3.01~5.65
12	Amylopectin	*qALP4-2*	2021A	Ft4	116.61	3.29	6.76	−1.77	112.4~117.7	Block8396~Block8758	34.64~40.42
13	Amylopectin	*qALP4-3*	2021S	Ft4	122.11	4.72	8.59	−1.28	120.3~126.9	Block8752~Block8836	39.85~42.65
			2022S	Ft4	123.21	7.37	14.41	−2.28			
			BLUP	Ft4	123.21	6.44	10.16	−0.19			
14	Amylopectin	*qALP7-1*	2021A	Ft7	58.11	2.62	5.53	0.68	47.8~61.4	Block13692~Block13766	12.1~15.32
15	Amylopectin	*qALP7-2*	2021S	Ft7	120.31	3.61	6.8	1.37	116.7~121.3	Block14329~Block14351	37.07~38.89
16	Amylopectin	*qALP7-3*	BLUP	Ft7	161.01	2.74	4.07	0.12	152.7~164.8	Block14552~Block14603	44.53~46.04
17	Total Starch	*qTSC1-1*	2022S	Ft1	39.81	2.91	5.76	−1.11	38.9~44.6	Block334~Block412	6.45~8.72
18	Total Starch	*qTSC1-2*	BLUP	Ft1	152.71	4.33	7.45	0.04	150.4~154.8	Block709~Block714	44.85~45.09
19	Total Starch	*qTSC4-1*	2021S	Ft4	194.81	3.01	5.98	−1.12	193.1~195	Block9392~Block9381	55.13~56.65
20	Total Starch	*qTSC7-1*	BLUP	Ft7	119.01	2.54	4.41	0.03	113.3~124.9	Block14276~Block14405	35.85~40.63
			2021A	Ft7	120.81	3.9	7.99	1.42			

R^2^ represents phenotypic variation that the QTL could explain. Positive and negative values in additive effect represent favorable alleles from Jinqiaomai2 and Xiaomiqiao, respectively.

**Table 4 ijms-25-09243-t004:** QTL clusters detected in the XJ-RILs population in four environments.

NO.	Cluster Name	Chromosome	Confidence Interval (cM)	Flanking Markers	Physical Interval (Mbp)	QTL Number	QTL
1	qClu-1-1	Ft1	35.7~46.3	Block320~Block428	6.45~9.15	3	*qALP1-1*, *qTSC1-1*, *qALP1-2*
2	qClu-4-1	Ft4	120.3~136.5	Block8744~Block8844	39.85~43.34	3	*qALP4-3**, *qAL4-1#*, *qAL4-2**
3	qClu-7-1	Ft7	113.3~124.9	Block14276~Block14405	35.85~40.63	2	*qTSC7-1#*, *qALP7-2*

* ^#^ indicate QTLs repeatedly detected in three and two environments, respectively.

**Table 5 ijms-25-09243-t005:** SNP/InDel variations identified in candidate genes with higher expression in seeds.

NO.	Name	Chr	Orientation	Times of Mutation	Mutation Type	Expression Cluster	Mutation Region	Rice Annotation
1	FtPinG0004897000.01	Ft4	−	14	SNP/Indel	C5	5’ UTR/3’ UTR	LOC_Os12g05430|ribosomal protein L24, putative, expressed
2	FtPinG0004897100.01	Ft4	+	6	SNP/Indel	C6	5’ UTR/3’ UTR	LOC_Os07g39820|SHR, putative, expressed
3	FtPinG0004899000.01	Ft4	−	1	SNP	C6	5’ UTR	LOC_Os03g31690|GCN5-related N-acetyltransferase, putative, expressed
4	FtPinG0002636200.01	Ft4	+	1	SNP	C5	3’ UTR	LOC_Os02g49140|glycosyltransferase, putative, expressed
5	FtPinG0002635900.01	Ft4	−	1	Indel	C5	3’ UTR	LOC_Os03g51990|ACT domain-containing protein, expressed
6	FtPinG0008593800.01	Ft4	+	2	SNP	C5	exon/3’ UTR	LOC_Os07g41570|expressed protein
7	FtPinG0008592400.01	Ft4	+	6	SNP/Indel	C5	5’ UTR/3’ UTR	LOC_Os03g07570|aminotransferase, putative, expressed
8	FtPinG0007101700.01	Ft4	+	1	Indel	C6	3’ UTR	LOC_Os08g33210|expressed protein
9	FtPinG0005926000.01	Ft4	+	4	SNP/Indel	C4	3’ UTR	LOC_Os03g60750|ribosomal RNA large subunit methyltransferase J, putative, expressed
10	FtPinG0005925700.01	Ft4	−	2	Indel	C5	3’ UTR	LOC_Os01g74146|WD repeat-containing protein, putative, expressed
11	FtPinG0009329200.01	Ft4	+	1	SNP	C5	intron	LOC_Os03g17740|transporter, putative, expressed
12	FtPinG0006225500.01	Ft4	−	1	SNP	C6	3’ UTR	LOC_Os03g60720|expansin precursor, putative, expressed
13	FtPinG0008469200.01	Ft4	+	1	SNP	C5	5’ UTR	LOC_Os03g52320|GRF-interacting factor 1, putative, expressed
14	FtPinG0008468900.01	Ft4	−	1	SNP	C5	intron	LOC_Os03g52320|GRF-interacting factor 1, putative, expressed
15	FtPinG0007371600.01	Ft4	+	1	SNP	C5	3’ UTR	LOC_Os03g49510|phosphatidylinositol-4-phosphate 5-kinase, putative, expressed
16	FtPinG0007371400.01	Ft4	+	13	SNP	C6	5’ UTR/3’ UTR	LOC_Os09g28110|hydroxyproline-rich glycoprotein family protein, putative, expressed
17	FtPinG0006002900.01	Ft4	+	1	SNP	C4	5’ UTR	LOC_Os12g03260|MATE efflux family protein, putative, expressed
18	FtPinG0005124100.01	Ft4	−	1	SNP	C5	5’ UTR	LOC_Os03g28940|ZIM domain-containing protein, putative, expressed
19	FtPinG0005109900.01	Ft4	−	1	SNP	C5	3’ UTR	LOC_Os02g51070|starch synthase, putative, expressed

## Data Availability

The genomic data of Tartary buckwheat are available from http://www.mbkbase.org/Pinku1/, accessed on 1 August 2024. The raw sequence data of transcriptome have been deposited in the Genome Sequence Archive, which is publicly accessible at http://download.big.ac.cn/gsa2/CRA008415/, accessed on 1 August 2024. The accession numbers for Xiaomiqiao were CRR577341, CRR577342, and CRR577343. The accession number for Jinqiaomai2 were CRR577359, CRR577360, and CRR577361.

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
