# Peer review of "QTL Mapping and Candidate Gene Analysis for Starch-Related Traits in Tartary Buckwheat (Fagopyrum tataricum (L.) Gaertn)"

_ijms, 2024, doi:10.3390/ijms25179243_

Round 1

Reviewer 1 Report

Comments and Suggestions for Authors

In the paper titled " QTL mapping and candidate gene analysis for starch related traits in Tartary buckwheat (Fagopyrum tataricum)", the authors carried out mapping analysis for starch related QTL and speculated that the five key candidate genes as its candidate gene. An in-depth analysis of the key candidate genes was conducted, including aspects such as annotation, expression patterns, and SNP/InDel variations, providing a foundation for further research. Findings in this study laid a foundation for understanding the molecular mechanism of starch, and for breeding cotton varieties with higher starch. My detailed comments are as follows:

1. The functional verification of the candidate genes is relatively preliminary, only verified through expression analysis and some SNP/InDel variations. There is a lack of more in-depth functional studies. It is suggested to supplement the relevant research progress in the discussion section.

2. Combining biochemistry and molecular biology techniques to study the specific role of the protein encoded by the candidate gene in the starch synthesis pathway.

3. Conduct a more in-depth discussion on the application prospects of the research results to provide more specific suggestions for molecular breeding of tartary buckwheat.

Comments on the Quality of English Language

 Minor editing of English language required.

Reviewer 2 Report

Comments and Suggestions for Authors

The manuscript is prepared on an interesting and current topic, which is significant from the point of view of genetics and plant breeding. Before accepting it, it is necessary to make certain additions and clarifications:   

Keywords - there is a lot of overlap with the title of the manuscript, which should not be. It needs to be adjusted.   

Introduction - is well-written and presents the issue appropriately, although I would venture to suggest the involvement of the current article (DOI: 10.17221/16/2023-CJGPB) on transcriptomic profiling in S. bicolor related to the content of starch and phenolic substances during grain development (no, it might be better to apply this article more in the discussion). From a formal point of view, TFs as part of the genome should be written in italics (line 83) and similarly GBSS (107) and SSS (108).   

Results - the title of chapter 2.1 is misleading. If the authors use the word "phenotype", then many imagine a morphological description of plants, but here it is an interpretation of the analysis of the content of substances, so I would change the name of this subsection so that it is not misleading and clearly declares the content of the text. Figure 3B - missing description why part of the text in red and part of the text in black. It is appropriate to state this because the references to Fig./Tab. they are somewhere more distant from each other in the text. Figure 5 - what kind of variability do the ears show? What about asterisks? Indeed, there is no statistically significant difference in the content of Fig. 5A?   

Discussion - is written legibly. The question is whether, in the case that the work is focused on starch in grain, it is appropriate to use references to potatoes or sweet potatoes and not emphasize studies focused on grain within cereals and pseudocereals. Alternatively, it is necessary to elaborate more on why?    Materials and Methods - subchapter 4.2 has a similarly misleading title as subchapter 2.1 - should be edited. In the Material and Methods section, any differences between the periods (2021S, 2021A, and 2022S) should be described or declared that external factors could not play a significant role for the starch content results. According to the results, there are certain differences in the period.  

References - in the case of magazine names, the standard in writing their names is not respected, i.e. few vs. upper case.

Round 2

Reviewer 2 Report

Comments and Suggestions for Authors

The authors accepted all my observations and comments. I recommend accepting the manuscript.